# Registered Clinical Trials for Artificial Intelligence in Lung Disease: A Scoping Review on ClinicalTrials.gov

**DOI:** 10.3390/diagnostics12123046

**Published:** 2022-12-05

**Authors:** Bingjie Li, Lisha Jiang, Dan Lin, Jingsi Dong

**Affiliations:** 1Department of Thoracic Oncology and State Key Laboratory of Biotherapy, Cancer Center, West China Hospital, Sichuan University, Chengdu 610017, China; 2Department of Nuclear Medicine, West China Hospital, Sichuan University, Chengdu 610017, China; 3Lung Cancer Center, West China Hospital, Sichuan University, Chengdu 610017, China

**Keywords:** artificial intelligence, machine learning, deep learning, lung disease, clinical study, ClinicalTrials.gov

## Abstract

Clinical trials are the most effective tools to evaluate the advantages of various diagnostic and treatment modalities. AI used in medical issues, including screening, diagnosis, and treatment decisions, improves health outcomes and patient experiences. This study’s objective was to investigate the traits of registered trials on artificial intelligence for lung disease. Clinical studies on AI for lung disease that were present in the ClinicalTrials.gov database were searched, and fifty-three registered trials were included. Forty-six (72.1%) were observational trials, compared to seven (27.9%) that were interventional trials. Only eight trials (15.4%) were completed. Thirty (56.6%) trials were accepting applicants. Clinical studies often included a large number of cases; for example, 24 (32.0%) trials included samples of 100–1000 cases, while 14 (17.5%) trials included samples of 1000–2000 cases. Of the interventional trials, twenty (15.7%) were retrospective studies and twenty (65.7%) were prospective studies.

## 1. Introduction

Deep learning, a branch of machine learning built on artificial neural networks, has displaced computers as the most capable tool for automating repetitive tasks [1,2] Additionally, deep learning can provide machine intelligence by imitating features of the human brain through repeated data training and algorithm iterations, or, to put it another way, by enabling a machine to process data for analysis and action, similar to a pro [3,4]. It can assist medical professionals during ordinary clinical workflows with its quick and reliable object identification, segmentation, tracking, and categorization of pathologic anatomical structures. AI has been utilized in a variety of medical situations, including screening [5,6], diagnosis [7], differential diagnosis [8], therapy, and treatment decisions [9], to increase efficiency and reduce costs. Additionally, research on the use of AI in imaging, particularly in the diagnosis and grading of COVID-19, is growing [10].

Chronic obstructive pulmonary disease (COPD) and asthma are the two most prevalent chronic lung diseases, but other conditions include pneumoconiosis, interstitial lung disease (ILD), pulmonary embolism (PE), and pulmonary hypertension, which are also global public health concerns. Globally, lung illnesses impact millions of individuals and result in high economic costs. AI can diagnose cancer and pneumonia using pathology and imaging just as well as trained medical experts. The production of vast quantities of health–disease data has accelerated via medical operations in the big data era. Doctors may provide patient care more effectively and precisely thanks to AI [1,11,12]. Because trials are the most trustworthy way to assess the advantages of diagnostic and treatment approaches, numerous researchers have used them to explore AI for lung disease. None of the published studies have compared the AI applications in lung disease of clinical trial data from ClinicalTrials.gov. This study presents the first scoping review of AI applications in lung disease.

## 2. Materials and Methods

### 2.1. Search Technique

ClinicalTrials.gov was used in May 2022 to conduct a thorough literature search on clinical studies conducted between 1 January 2014 and 3 May 2022 following the procedure in Figure 1, using keywords such as pulmonary disease, lung disease, machine learning, deep learning, and artificial intelligence. As we know, AI is a rather big concept containing serial subset concepts, including machine learning and deep learning. Clinical doctors do not describe their unique algorithms in detail on ClinicalTrials.gov. Three different researchers searched the database to discover articles describing the use of AI on lung illness for varied purposes, i.e., (1) comparing radiologist and AI algorithm reading abilities of thoracic CT scans; (2) employing devices to gather symptoms and make clinical decisions; and (3) identifying patients for specific treatments. All of the articles were chosen manually.

### 2.2. Selection Standards

Registered trials using AI methodology (such as ML or DL) for lung illness met the inclusion criteria (e.g., lung cancer, PNs, or COVID-19) including screening, diagnosis, prognosis assessment, or outcome prediction. Incomplete registration information was a criterion for disqualification.

### 2.3. Data Extraction and Trial Screening

Based on the study titles, the authors read the designs of clinical trials, including study details, tabular views, and posted results. There were discussions if there were any differences of opinion. Two writers then independently extracted the data from the included trials. The following information was extracted: registered number, study type, first submitted date, first posted date, the last update posted date, actual study start date, current primary outcome measures, current secondary outcome measures, brief title, study type, study design, target follow-up duration, study population, condition, intervention, study groups, publications, recruitment status, estimated enrollment, eligibility criteria, phases of the trial, allocation, intervention model, masking and intervention, and types of disease, sex, ages, listed location countries, NCT number, study sponsor, collaborators, investigators, and verification date.

### 2.4. Data Evaluation

The study’s technique is comparable to that of our earlier investigation [13]. The outcomes included year, trial status, study results, age, sex, enrolment, sponsor, allocation, sponsors, study design parameters, disease kinds, and application method.

## 3. Results

### 3.1. The Features of the Tests Included

Three hundred and forty-four results from the ClinicalTrials.gov website were found. After carefully reviewing the clinical study data, two hundred and ninety-one outcomes were excluded. Finally, there were fifty-three trials in total for this investigation (Appendix A). Similar to our earlier study, forty-six trials (86.8%) were observational, whereas just seven (13.2%) were interventional trials. Eighteen trials remained unrecruited, with the majority of trials still in the recruitment stage. There were fifteen (15.4%) completed studies, but the trial outcomes were not available. Three clinical trials had age limitations, i.e., only patients over the age of eighteen were enrolled in the studies (5.7%), while fifty trials included participants of all ages (94.3%). The studies often featured a large number of participants; 24 (45.3%) trials consisted of samples from 100 to 1000 cases, 14 (26.4%) trials included samples from 1001 to 2000 cases, and 8 (15%) trials included samples from more than 2000 cases. The primary sponsors of fifteen studies (28.3%), nineteen trials (35.6%), and nineteen trials (35.8%) were cited as universities, hospitals, and corporations, respectively (35.8%). The majority of the trials—27—were undertaken in Europe, with the remainder—11 in North America, 11 in Asia, and 4 in Australia. The features of the trials are listed in Table 1.

### 3.2. Features of the Research Design

The stages of other trials were irrelevant, and just two of the twenty-seven interventional experiments were in phase 1/2. Only one (14.3%) of the interventional studies served both diagnostic and therapeutic objectives; the majority (85.7%) were for diagnoses. Two of the trials were non-randomized, and five trials (71.4%) failed to correctly determine the allocation value (28.6%). There were four assignments in a single group, two parallel assignments (28.6%), one sequential assignment (14.3%), and two parallel assignments (57.1%). Seven interventional studies were conducted, and six (85.7%) of them were conducted without masking, as opposed to one (14.3%) with masking. The features of the interventional trials are presented in Table 2.

Twenty-eight trials (60.9%) of the observational trials were cohort studies, six (13.1%) were case-only trials, three (6.5%) were case-control trials, one (2.2%) was a case-crossover trial, and seven (15.2%) were other trials. Out of all the observational trials, twenty-two (47.8%) were prospective studies, twenty (43.5%) were retrospective studies, one (2.2%) was cross-sectional, and three (6.5%) were other types of trials. The traits of the observational trials (n = 46) are displayed in Table 3.

### 3.3. Overview of Diagnostic Clinical Trials

From the fifty-three studies, eighteen (34.0%) were for lung cancer, sixteen (30.2%) were for COVID-19 and other infectious lung illnesses, five (9.4%) were for COPD/ILD, and one (1.9%) was for IIP, as well as trials for asthma, pneumoconiosis, osteosarcoma, pulmonary embolism, and pulmonary hypertension. Seventeen (32.1%) of these trials were for imaging diagnosis, followed by six (11.3%) for device diagnosis, five (9.4%) for biomarker diagnosis, and four (7.5%) for pathology diagnosis. The summary of trials for diagnosis is shown in Table 4.

## 4. Discussion

With deep neural networks, artificial intelligence has expanded significantly and quickly in recent years. This study offered a scoping review of the clinical trials on the use of AI in lung disease that were registered on ClinicalTrials.gov. Only a small number of trials were interventional, as our prior study showed [13], while the majority were observational. The majority of experiments were reported after 2018, proving that the use of AI in clinical settings was a recent development. In particular, the trials on COVID-19 based on the deep learning algorithm showed that it was necessary to broaden COVID-19 mass screening and identify patients at earlier stages of the disease [5,14,15,16]. There were no trials with available results, although eight of them were completed. Most trials tended to be large sample size studies, especially those trials related to COVID-19. Most trials were sponsored by hospitals and companies. Notably, the vast majority (n = 27) of trials were conducted in Europe.

There were eighteen trials designed to study lung cancer and sixteen trials designed to study COVID-19 and other infectious pulmonary diseases, which showed a new trend of the deep learning algorithm in differentiating and grading different images or clinical data. The artificial intelligence algorithm has proven to outperform physicians in discriminating respiratory pathologies via respiratory functional explorations, symptoms [17], radiological examinations [18,19], and pathological differentiation [20,21]. Artificial intelligence and machine learning-based automated diagnostic systems are being developed quickly, and they have the potential to boost diagnostic speed and accuracy while protecting doctors by limiting their contact with COVID-19 patients.

Song et al. reported that CT imaging features might be used to predict the pathological type of micropapillary adenocarcinoma and combining imaging parameters with clinical features can provide added diagnostic value to identify the presence of a micropapillary component [22]. In 2019, Google AI and collaborators published results obtained using a deep learning model for the prediction of lung cancer from CT scans. The authors claimed that their model outperformed radiologists when a single CT scan was analyzed and performed similarly to radiologists when a prior scan was also available [23]. It was anticipated that AI would provide more information for an accurate diagnosis. It was reported that machine learning could predict the histological type of lung cancer through the imaging characteristics of PET/CT. Luo et al. and Yu et al. reported that the automatic analysis method of AI could perform a pathological image analysis to predict the prognosis of patients with lung cancer [24,25]. The CT diagnosis and pathological diagnosis of lung cancer are important prerequisites for the treatment of lung cancer [25,26,27,28].

Chen et al. improved a new diagnosis method by combining the dual effects of the two algorithms (convolutional neural network (CNN) and the recurrent neural network (RNN)) to process the classification of benign and malignant nodules [20]. The AUC area for judging benign and malignant pulmonary nodules was 0.729 in the improved 3D U-net system. The AUC area of the radiologist group was 0.794. The improved 3D U-net system and radiologists have certain accuracies in judging benign and malignant pulmonary nodules. Niu et al. developed a deep-learning model to predict TMB based on histopathological images of LUAD (lung adenocarcinoma) whose AUC was 0.64 [29]. At this stage, artificial intelligence shows good performance in judging benign and malignant pulmonary nodules and has very promising application prospects. More advanced models of machine learning could improve accuracy, such as the recent models of classification used for other biological issues.

The reasons why deep learning algorithms were needed are as follows: Firstly, medicine is a time-acquired, experience-needed subject, based on doctors’ abilities to remember patterns from a previous time point or descriptions written in clinical notes, which means well-educated doctors with lifelong learning. Secondly, since the interpretation and reading of images are highly subjective skills, the accuracy of the results differs widely depending on experience and varies among different specialties. Thirdly, the heterogeneity of patients may originate from differences in the intrinsic properties of the device and extrinsic patient-related factors, such as obesity and patient compliance, so it takes a long time for doctors to make the right diagnoses. AI networks offer low-cost, reproducible, near-instantaneous classification options, which could be used in clinical settings. Additionally, in certain diseases, such tools could empower patients confined at home to self-monitor their symptoms to inform telemedicine officials and personalized care representatives.

During the recent pandemic crisis, researchers focused on analyzing appropriate COVID-19 diagnoses by implementing a convolutional neural network (CNN). Deep learning technology is a practical, valuable, and suitable technique that can be deemed reliable for adequate diagnosis of the COVID-19 virus. Various researchers have recently carried out the application of deep learning for COVID-19. CTs and X-rays have the most critical roles in medical imaging for diagnosing COVID-19. CTs have demonstrated high sensitivity and repeatability in the general diagnosis of COVID-19, body imaging [30], and in the ability to detect various types of opacities, e.g., ground-glass opacity (GGO), consolidation [31], which is mainly seen in COVID-19 patients. By analyzing CT scan images, Kogilavani et al. demonstrated the classification of CT images between COVID-19 and non-COVID-19. The authors used six deep learning architectures, namely, VGG16, DenseNet, MobileNet, Xception, NASNet, and EfficientNet. Among these models. The highest accuracy acquired among all the models was VGG16 at 97.68%. Hence, the proposed system identifies the VGG16 model as the best model to classify the given CT scan images into COVID and non-COVID. [14]. Wang et al. proposed a 3D deep convolutional neural network to detect COVID-19 from CT volumes; this algorithm obtained 0.959 ROC AUC and 0.976 PR AUC taking only 1.93 s to process a single patient’s CT. The developed deep learning software is available at: https://github.com/sydney0zq/covid-19-detection (accessed on 19 September 2022). Cai et al. used the UNet model for lung and lesion segmentation, which resulted in a dice similarity of 0.77 using a ten-fold CV protocol with a database of 250 pictures taken from 99 patients [32]. Saood et al. presented a COVID-19-based CT lung image segmentation system that used 100 COVID-19 lung CT images. The scientists evaluated the findings of the two models, UNet and SegNet, and found they had similar DS scores of 0.73 and 0.74, respectively [15]. The authors did not compare lung area errors or create JI or BA plots, and further, the authors did not apply PAI. Agarwal et al. designed COVLIAS 2.0 when refining eight different strategies for COVID-19-based CT lung segmentations with a storage reduction of ~97% and overall system error under 6%; thus, COVLIAS 2.0 is reliable, accurate, and stable in clinical settings [5].

This study had limitations. First, not all trials were included because we only looked for findings on ClinicalTrials.gov. Our analysis nevertheless managed to include the bulk of the studies because ClinicalTrials.gov consisted of more than 80% of those on the International Clinical Trials Registry Platform of the World Health Organization. To conduct a thorough investigation, we also looked through other clinical trial websites, such as SwissEthics. Second, the majority of the trials were completed recently, and ClinicalTrials.gov did not have any findings available. ClinicalTrials.gov did not require the doctors to elaborate on the algorithm, masking, intervention, etc., in detail, which caused a lot of misunderstanding. Third, although only fifty-three papers used AI in lung disease, not all specific lung diseases were carefully looked at.

## 5. Conclusions

In conclusion, the current analysis lists the basic characteristics of registered trials using AI on lung disease in preclinical and clinical research.

## Figures and Tables

**Figure 1 diagnostics-12-03046-f001:**
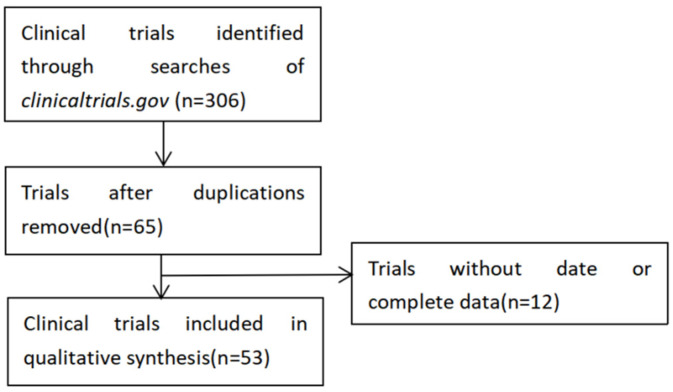
Flowchart of the selection process of the included studies.

**Table 1 diagnostics-12-03046-t001:** Characteristics of included trials.

Characteristics		Number	Percentage (%)
Study type			
	Interventional	7	13.2
	Observational	46	86.8
Year			
	2015–2019	16	30.2
	2019–2020	17	32.1
	2020–2021	12	22.6
	2021–2022	8	15.1
Status			
	Completed	8	15.1
	Recruiting	30	56.6
	Active, not recruiting	6	11.3
	Not yet recruiting	5	9.4
	Unknown status	4	7.5
	Withdrawn	0	0
Study results			
	Has available results	0	0
	No available results	53	100
Participant age (y)			
	1 to older	50	94.3
	Older than 18	3	5.7
Enrollment			
	<100	7	13.2
	100–1000	24	45.3
	1001–2000	14	26.4
	>2000	8	15.1
Sponsor			
	University	15	28.3
	Hospital	19	35.8
	Company	19	35.8
Location			
	Europe	27	50.9
	North America	11	20.8
	Asia	11	20.8
	Australia	4	7.5

**Table 2 diagnostics-12-03046-t002:** The characteristics of the interventional trials.

Characteristics		Number	Percentage
Primary purpose			
	Diagnostic and treatment	1	14.3
	Diagnostic	6	85.7
	Screening	0	0
Phase			
	Phase 1/2	2	28.6
	Phase 3/4	0	0
	Not applicable	5	71.4
Allocation			
	Randomized	1	14.3
	Non-randomized	2	28.6
	Missing value	4	57.1
Intervention model			
	Parallel assignment	2	28.6
	Sequential assignment	1	14.3
	Crossover assignment	0	0
	Single group assignment	4	57.1
Masking			
	Single	1	14.3
	Double	1	14.3
	Triple	0	0
	Without	5	71.4

**Table 3 diagnostics-12-03046-t003:** Study design elements of observational trials.

Characteristics		Number	Percentage (%)
Observational model			
	Case-only Case-control Case-crossover	6 3 1	13.0 6.5 2.2
	Cohort	28	60.9
	Ecological or community	1	2.2
	Other	7	15.2
Time perspective			
	Prospective	22	47.8
	Retrospective	20	43.5
	Cross-sectional	1	2.2
	Other	3	6.5

**Table 4 diagnostics-12-03046-t004:** Overview of clinical trials in diagnosis.

Characteristics		Number	Percentage (%)
Disease			
	Lung cancer	18	34.0
	COVID-19 and other infectious diseases	16	30.2
	COPD/ILD	5	9.4
	IIP/IPF/NSIF Asthma Pneumoconiosis	1 1 1	1.9 1.9 1.9
	Osteosarcoma	1	1.9
	Pulmonary embolism	1	1.9
	Pulmonary hypertension	1	1.9
	Non-specific	8	15.1
Application method			
	Device	6	11.3
	Endoscopy	1	1.9
	Imaging	17	32.1
	Pathology	4	7.5
	Biomarker	5	9.4
	Other	20	37.7

## Data Availability

Not applicable.

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
