# Peer review of "Registered Clinical Trials for Artificial Intelligence in Lung Disease: A Scoping Review on ClinicalTrials.gov"

_diagnostics, 2022, doi:10.3390/diagnostics12123046_

Round 1

Reviewer 1 Report

Primary observations: 

• The paper is written in completely vague manner, and does not contribute in any way. The study does not explain the relation of artificial intelligence and clinical trials. Nothing has been discussed regarding this. Only a fraction of it has been discussed in the discussion section. The article needs significant improvement and need clear writing too. 

1. The resources provided by these data can be used to study artificial intelligence (AI) in relation to various medical issues.

-What do you mean by studying artificial intelligence? This is an ambiguous statement. 

2. The objective of this study was to analyze the characteristics of registered trials on AI, deep learning, machine learning

-What do you mean by AI Here ? Either deep learning or Machine learning or ?

3. Only eight trials, or 15.4%, were completed.

-Vague statement 

4. Abstract is vague, and unclear, The title of the paper itself is not truly representing the content itself. 

5. There were different combinations of the terms pulmonary disease, lung disease, machine learning, deep learning and artificial intelligence.

-What do you mean by this?

Result section: 

Similar to our earlier study, 46 trials (86.8%), 96 were observational, whereas just 7 (13.2%) were interventional trials.

-What was the previous study? Not referred

Three patients over the age of 18 were enrolled in studies (5.7%) while fifty trials included participants of all ages (94.3 % ).

- Does all age, does not include age of 18?

Especially, the trials based on deep learning algorithm about COVID-19 showed that it was necessary to broaden COVID-19 146 mass screening and identify patients at earlier stages of disease[5, 14-16]. 

-How ?

Author Response

Dear reviewers, We appreciate the in-depth critiques and insightful suggestions, which have helped us greatly to raise the caliber of the work. We have carefully revised the article in light of your comments and recommendations, and we have addressed all of your issues. The "Manuscript with modifications monitored (in yellow)" contains the specific changes, and the point-by-point answer is as follows (reply in blue).

POINT-BY-POINT RESPONSE TO REVIEWER COMMENTS (reply in blue)

Reviewer#1

Primary observations: 

The paper is written in completely vague manner, and does not contribute in any way. The study does not explain the relation of artificial intelligence and clinical trials. Nothing has been discussed regarding this. Only a fraction of it has been discussed in the discussion section. The article needs significant improvement and need clear writing too. 

We really appreciated reviewers’ enthusiastic endorsement and helpful criticism. We have made the necessary adjustments and taken appropriate action in response to any concerns.

  1. The resources provided by these data can be used to study artificial intelligence (AI) in relation to various medical issues.

-What do you mean by studying artificial intelligence? This is an ambiguous statement. 

Thanks for your important questions. Sorry for using inappropriate and misleading statement. We wanted to show the potential of websites like ClinicalTrials.gov to illustrate the current scientific research of AI in various medical issues.

  1. The objective of this study was to analyze the characteristics of registered trials on AI, deep learning, machine learning

-What do you mean by AI Here ? Either deep learning or Machine learning or ?

We appreciate your thoughtful inquiries. As we known, AI is rather big concept containing serial subset concept including machine learning and deep learning. Clinical doctors don't really need to describe their unique algorithms in detail for their studies on a clinical trials website. By searching the ClinicalTrials.gov, we found that the clinical doctors used these three words to express using certain algorithms capable of looking at vast amounts of data, so we used these as our keywords. We change it in appropriate statement.  

  1. Only eight trials, or 15.4%, were completed.

-Vague statement 

Thanks for your suggestion. We are sorry for the misleading sentence. We have modified the sentence you stated and cited the paper in the new draft. Thank you for reminding.

  1. Abstract is vague, and unclear, The title of the paper itself is not truly representing the content itself. 

Thanks for these insightful comments. We added more detailed information and revised our manuscript.  

  1. There were different combinations of the terms pulmonary disease, lung disease, machine learning, deep learning and artificial intelligence.

-What do you mean by this?

Thanks for your helpful suggestions. We modified our statement in appropriate way.

Result section: 

Similar to our earlier study, 46 trials (86.8%), 96 were observational, whereas just 7 (13.2%) were interventional trials.

-What was the previous study? Not referred

We are grateful to your suggestion and revised statement in our manuscript.

Three patients over the age of 18 were enrolled in studies (5.7%) while fifty trials included participants of all ages (94.3 % ).

- Does all age, does not include age of 18?

Thank you for the insightful advice. Some clinical studies enrolled patients of any age, while others did not. We classified them as "participants of all ages" in those clinical studies that did not have an age limitation.

Especially, the trials based on deep learning algorithm about COVID-19 showed that it was necessary to broaden COVID-19 146 mass screening and identify patients at earlier stages of disease[5, 14-16]. 

-How ?

 We appreciate your question. Chest CT scans have demonstrated great sensitivity in identifying COVID-19 in some studies, and it took a long time for radiologists to read these massive CT scans. In order to accurately distinguish between viral pneumonia (COVID-19) and bacterial pneumonia (COVID), AI systems could be applied to quickly screen and diagnose a large number of patients with suspected COVID-19 and decrease unnecessary delays.

Reviewer 2 Report

The contribution and motivation are not clear in this study

where the comparison study with other Artificial Intelligence

where the protocol that used in this study. 

Author Response

Dear reviewer, We appreciate the in-depth critiques and insightful suggestions, which have helped us greatly to raise the caliber of the work. We have carefully revised the article in light of your comments and recommendations, and we have addressed all of your issues. The "Manuscript with modifications monitored (in yellow)" contains the specific changes, and the point-by-point answer is as follows (reply in blue).

Reviewer#2

The contribution and motivation are not clear in this study

Thank you for this valuable feedback.This study's primary purpose is to comprehend the current state of AI application in the study of lung disease, which may assist the researchers by providing related studies in a methodical way. Our research confirms the findings of the current study presenting the characteristics of registered trials on AI for lung disease especially the COVID-19. We have added a sentence to the Disscusion section to clarify this.

where the comparison study with other Artificial Intelligence

We appreciate reviewers for this kind recommendation. Clinical doctors don't elaborate their algorithms point for point on a clinical trials website as a result, we failed to compare study with other Artificial Intelligence one by one. Thank you for your sincere advise and we have added sentence to the Discussion section to compare the Artificial Intelligence algorithms in published articles.

where the protocol that used in this study. 

Thank you for this valuable feedback. We have shown our whole protocol that used in this study in our Materials and Methods part. Three different researchers searched ClinicalTrials.gov to find articles describing the use of AI on lung illness using keywords like pulmonary disease, lung disease, machine learning, deep learning and artificial intelligence from January 1, 2014 to May 3, 2022. All of the articles were chosen manually.

Reviewer 3 Report

• The primary output/endpoint variable(s)/measurement(s) of the study should be defined.  • Which randomization method was used in the distribution of the individuals included in the study to the groups? 

• Which blinding (masking) method was used in the study? 

• How was the sample size determined? This information should be explained in the Materials and Methods section. 

• Which sampling (probable or non-probable, etc.) method was used in the study?

Author Response

Dear reviewer, We appreciate the in-depth critiques and insightful suggestions, which have helped us greatly to raise the caliber of the work. We have carefully revised the article in light of your comments and recommendations, and we have addressed all of your issues. The "Manuscript with modifications monitored (in yellow)" contains the specific changes, and the point-by-point answer is as follows (reply in blue).

POINT-BY-POINT RESPONSE TO REVIEWER COMMENTS (reply in blue)

  • The primary output/endpoint variable(s)/measurement(s) of the study should be defined. 

Thank you for this valuable feedback.This study's endpoint variables were the information of included clinical trials including registered number, study type, first submitted date, first posted date, the last update posted date, actual study start date, current primary outcome measures, current secondary outcome measures, brief title, study type, study design, target follow-up duration, study population, condition, intervention, study groups, publications, recruitment status, estimated enrollment, eligibility criteria, phases of the trial, allocation, intervention model, masking and intervention, and types of disease, sex, ages, listed location countries, NCT number, study sponsor, collaborators, investigators, verification date. We have added the supplementary information.

  • Which randomization method was used in the distribution of the individuals included in the study to the groups? 

We appreciate reviewers for this kind recommendation. The randomization method used in different clinical trials varied from each other. In the clinical trial, NCT05221814, its entire dataset was divided into training and test datasets, which were mutually exclusive, using random sampling. The clinical doctors didn’t describe their randomization method with detail which we have added sentence about limitation about our study to the Discussion section.

  • Which blinding (masking) method was used in the study? 

Thanks for your insightful comments. The aim of our study is to mining of clinical trial data to grasp the application trend of AI in lung disease. Three different researchers searched ClinicalTrials.gov respectively to find the related information and it is key to ensure the objectivity of the evaluation by independent evaluation and cross-checking. In a clinical trial located in Pakistan(NCT05268263), which is studying sounds acquired from stethoscope, its gold standard is the label given to each lung sound recording by an experienced consultant pulmonologist. The AI model is blinded to these labels and is tested independently for detection of normal lung sounds, wheezes, and crackles. Likely, in another clinical trial(NCT03487952) evaluating lung nodule, the diagnosis process was conducted via double-blind method and detection rates of AI and radiologist will be recorded respectively. The clinical doctors didn’t describe their masking method with detail which we have added sentence about limitation about our study to the Discussion section.

  • How was the sample size determined? This information should be explained in the Materials and Methods section. 

Thank you for this valuable feedback. We have to admit we didn’t determine the specific sample size of our study at first until searching ClinicalTrials.gov and screening the clinical trials. 

  • Which sampling (probable or non-probable, etc.) method was used in the study?

Thank you for your question. Our study is a scoping review on ClinicalTrials.gov that helps to determine the characteristics of all the registered trials on AI for lung disease between January 1, 2014 and May 3, 2022. Our data and analysis were based on all the samples collected by the procedure.

Round 2

Reviewer 1 Report

All of the concerned areas are addressed. Thank you.

Reviewer 2 Report

accept in current form only english still needs improvements.